# Persistent post-discharge opioid prescribing after traumatic brain injury requiring intensive care unit admission: A cross-sectional study with longitudinal outcome

Lauren K. Dunn[ID][1]\*, Davis G. Taylor[2], Samantha J Smith[1], Alexander J. Skojec[ID][1], Tony R. Wang[2], Joyce Chung[1], Mark F. Hanak[ID][1], Christopher D. Lacomis[1], Justin D. Palmer[1], Caroline Ruminski[1], Shenghao Fang[ID][1], Siny Tsang[ID][3], Sarah N. Spangler[1], Marcel E. Durieux[1,2], Bhiken I. Naik[1,2]

1 Department of Anesthesiology, University of Virginia Health System, Charlottesville, Virginia, United States of America, 2 Department of Neurological Surgery, University of Virginia Health System, Charlottesville, Virginia, United States of America, 3 Nutrition and Exercise Physiology, Washington State University, Spokane, Washington, United States of America

\* lak3r@virginia.edu

**Data Availability Statement:** All relevant data are within the paper and its Supporting Information files.

## Abstract

Traumatic brain injury (TBI) is associated with increased risk for psychological and substance use disorders. The study aim is to determine incidence and risk factors for persistent opioid prescription after hospitalization for TBI. Electronic medical records of patients age ≥ 18 admitted to a neuroscience intensive care unit between January 2013 and February 2017 for an intracranial injury were retrospectively reviewed. Primary outcome was opioid use through 12 months post-hospital discharge. A total of 298 patients with complete data were included in the analysis. The prevalence of opioid use among preadmission opioid users was 48 (87%), 36 (69%) and 22 (56%) at 1, 6 and 12-months post-discharge, respectively. In the opioid naïve group, 69 (41%), 24 (23%) and 17 (19%) were prescribed opioids at 1, 6 and 12 months, respectively. Preadmission opioid use (OR 324.8, 95% CI 23.1–16907.5, p = 0.0004) and higher opioid requirements during hospitalization (OR 4.5, 95% CI 1.8–16.3, p = 0.006) were independently associated with an increased risk of being prescribed opioids 12 months post-discharge. These factors may be used to identify and target at-risk patients for intervention.

## Introduction

Traumatic brain injury is a significant global and national public health concern. According to the Global Burden of Diseases, Injuries, and Risk Factors Study, an estimated 27 million new cases of TBI occurred in 2016.[1] In the United States, the Centers for Disease Control reported an estimated 2.8 million TBI-related emergency department visits, hospitalizations, and deaths in 2013 with an estimated direct and indirect economic cost of these injuries of approximately $76.5 billion (based on 2010 data).[2] Although death rates from TBI have

**Funding:** The author(s) received no specific funding for this work.

**Competing interests:** The authors have declared that no competing interests exist.

declined by 5% between 2007 to 2013, TBI-related emergency department visits have increased by 47% over the same period [2], suggesting that more patients are living with TBI-related disability.[3, 4]

Patients with traumatic brain injury (TBI) are at increased risk for psychological disorders including suicide, post-traumatic stress and mood disorders, and for substance abuse; however, data regarding opioid use following TBI in the general population is lacking.[5] Opioids [6] together with benzodiazepines [7] are frequently used in the acute treatment of critically brain-injured patients for the management of intracranial hypertension, to provide analgesia and anxiolysis and to facilitate mechanical ventilation. Opioids and benzodiazepines reduce cerebral metabolic oxygen consumption with minimal effects on cerebral perfusion, which makes them ideal agents for the management of TBI.[8–10] However, exposure to opioids for the acute management of TBI may increase risk for dependence and opioid use disorder in these patients. We and others have shown that in-hospital exposure to opioids increases the risk for chronic opioid use in surgical patients [11–13]; with persistent use reported as much as one year after the index admission. Therefore, it is conceivable that in-hospital exposure to opioids might predispose patients with TBI to continue to use opioids after hospital discharge.

The aim of this study is to determine the incidence and risk factors for persistent opioid prescription (up to one year after admission) in patients with a primary TBI. We hypothesized that opioid use prior to hospitalization and in-hospital exposure to opioids for management of TBI would be associated with increased risk for persistent opioid prescription 1 year after hospital discharge.

## Material and methods

This manuscript adheres to the STROBE guidelines. The study was approved by the Institutional Review Board (IRB) for Health Science Research (HSR-19922) and the requirement for written informed consent was waived by the IRB. All human and animal studies have been approved by the institution ethics committee and have been performed in accordance with the ethical standards laid down in the 1964 Declaration of Helsinki and its later amendments.

This was a single-center retrospective study. ICD-10 diagnosis code, demographic data and the electronic medical record were used to identify patients age ≥ 18 who were admitted between January 2013 and February 2017 to our neuroscience intensive care unit (ICU) with a primary admission diagnosis of TBI. The neuroscience ICU cares for patients with isolated traumatic brain or spine injury. Patients with polytrauma are cared for in our Surgical Trauma ICU and were not included in our patient cohort. Surgical CPT codes were used to identify any surgical procedures that were performed during the index admission of the study cohort. Patients with major chest, abdomen, pelvis, vascular or long-bone fractures with an associated TBI were excluded from this study, to remove the confounding effect of non-neurological injuries on persistent post discharge opioid prescription. The primary outcome was persistent opioid prescription through 12 months after discharge, as defined by having a prescription for one or more opioid medications at each of 1, 6, and 12-month time points. Secondary outcomes were persistent opioid prescription at 1 and 6 months post-discharge, defined as prescription at 1 month and prescription at 1 *and* 6 months, respectively.

### Covariates

Patient demographic data (age, sex, ethnicity), weight, body mass index (BMI), psychiatric and substance abuse history, preadmission medication use including non-steroidal anti-inflammatory medications (NSAIDs), acetaminophen, benzodiazepines, muscle relaxants, antidepressants, antipsychotics, opioids and daily oral morphine-equivalent (OME) dose (for dose

conversion see http://www.uptodate.com/contents/cancerpain-management-with-opioids-optimizing-analgesia) [14] (complete medication list in S1 Table), in-hospital Verbal Response Scale (VRS) and Critical Care Pain Observation Tool (CPOT) pain scores, in hospital medication administrations, opioid administration 48 hours prior to discharge (yes/no), in-hospital intubation, surgical CPT codes, intensive care unit (ICU) and hospital length of stay (LOS), discharge disposition and prescription medication data from 1, 6 and 12 months after discharge were recorded. These covariates were chosen *a priori* based on previous studies that demonstrated an association with persistent opioid use after surgery (e.g. antidepressant use, substance abuse history), biological plausibility (e.g. in-hospital opioid exposure) and previously unexplored variables (e.g. opioid exposure 48 hrs prior to discharge and discharge disposition).

OME was calculated by converting and summating all enteral and parenteral opioid administered within a 24-hour period, which was then standardized to BMI (mg/kg/m$^2$). To control for duration of opioid exposure during the hospital stay the total daily OME was divided by length of hospital stay (days) and BMI.

A sensitivity analysis was performed in those patients who were taking opioids pro re nata (PRN) prior to their admission. Subjects were assumed to be taking either some (25%) or all (100%) of the PRN dose. For example, a subject with a prescription for "10 mg oxycodone every 4 hours PRN pain" was estimated to be taking between 25% of the prescribed amount (15 mg oxycodone) or 100% of the prescribed amount (60 mg oxycodone) daily.

### Data validation

To ensure data accuracy for all reported variables, we used the Microsoft Excel function "RANDBETWEEN" to generate 150 random numbers between the highest and lowest case numbers. Thereafter, we manually validated all data for the associated random generated case numbers using the hospital electronic medical record.

### Statistical analysis

Descriptive statistics are presented as number (*n*) and proportion (%) for dichotomous variables, and mean (*M*) and standard deviation (*SD*) for continuous variables.

Differences between patients using opioids prior to admission and opioid naïve patients were compared using Fisher/chi-square tests (for dichotomous or categorical variables), linear regression models (for continuous variables), and generalized linear regression models with a logit function (for count variables). Subjects with missing preadmission opioid use data were excluded from the analysis.

Generalized linear models were used to examine whether the odds of persistent opioid prescription 1, 6 or 12 months after discharge. Prior to all predictive modeling, collinearity among covariates was assessed using the variance inflation factor: if greater than 5, covariates were selectively excluded from the model. All analyses were performed in R version 3.3.2.[15]

## Results

### Patient demographics and preadmission characteristics

Of 528 records reviewed, preadmission opioid use data was missing for 134 subjects, 65 patients had no TBI on radiological imaging and 32 were readmissions following their TBI. A total of 297 subjects were included in the final analysis. Seventy-six were preadmission opioid users and 221 were opioid naïve prior to admission. A consort diagram is shown in Fig 1.

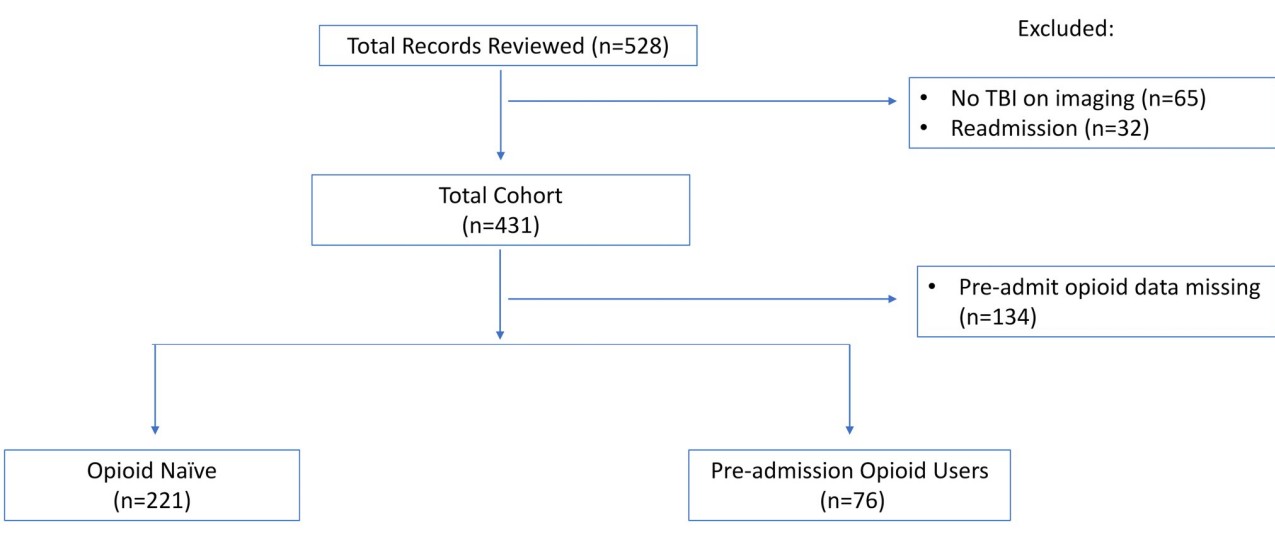

**Fig 1. Consolidated standards of Reporting trials (CONSORT) diagram.**

Patient demographics, preadmission characteristics and in-hospital exposures are shown in Table 1. A greater percentage of preadmission opioid users were women (p = 0.05). Preadmission history of depression (p < 0.001), use of acetaminophen (p < 0.001), benzodiazepines (p < 0.001), muscle relaxants (p = 0.009), and antidepressants (p = 0.013) were significantly higher among preadmission opioid users compared to opioid naïve patients. Among patients using opioids prior to admission, the mean daily OME dose was 22.8 ± 55.2 mg (minimum 0.00, maximum 407.5 mg) assuming 25% of the prescribed PRN dose and mean 28.1 ± 56.7 mg (minimum 0.0 mg, maximum 430.0 mg) assuming 100% of the prescribed PRN dose.

There were similar rates of intubation and surgery between groups. There were no significant differences in discharge disposition between groups.

Daily and weekly OME (per BMI), average daily OME (per BMI) are shown in S2 Table. VRS and CPOT pain scores were excluded from the analysis due to large amounts of missing data. Patients using opioids prior to admission received significantly more benzodiazepine (p = 0.002), muscle relaxant (p<0.001), antidepressant (p<0.001), and opioid (p = 0.003) administrations compared to opioid naïve patients. Sixty-three patients (83%) using opioids prior to admission received opioids in the 48 hours prior to discharge compared to 136 (62%) of previously opioid naïve patients, which was a statistically significant difference (p = 0.001).

The proportion of patients prescribed post-discharge opioids and other medications is shown in Table 2. Among preadmission opioid users the point prevalence of opioid prescription at 1, 6 and 12-month post-discharge was 48 (87%), 36 (69%) and 22 (56%) while among previously opioid naïve patients the prevalence rate was 69 (41%), 24 (23%) and 17 (19%). A greater number of preadmission opioid users had a prescription for opioids at 1, 6 and 12 months compared to opioid naïve patients at each of these time points (p<0.001). Preadmission opioid users were also prescribed more benzodiazepines at 1 (p = 0.009) months and more muscle relaxants at 1 (p<0.001) and 6 months (p = 0.046).

## Persistent opioid prescription after hospital discharge

The number of preadmission opioid users who were persistently prescribed opioids through 1, 6 and 12 months was 48 (87%), 30 (61%) and 18 (44%), respectively. Sixty-nine (42%), 15 (11%) and 3 (2%) previously opioid naïve patients were persistently prescribed opioids

**Table 1.** Comparison of patient demographics, preadmission characteristics and in-hospital exposures between preadmission opioid users and opioid naïve patients.

| Characteristic | Total (n = 431) | Preadmit Opioid Use (n = 76) | Opioid Naïve (n = 221) | p value |
|---|---|---|---|---|
| **Preadmission Characteristics** | | | | |
| Age (years) | 63 ± 20 | 64 ± 18 | 65 ± 19 | 0.64 |
| Sex | 263 (61%) Men | 37 (49%) Men | 139 (63%) Men | **0.05** |
| | 168 (39%) Women | 39 (51%) Women | 83 (37%) Women | |
| Ethnicity | 389 (90%) White | 68 (89%) White | 199 (90%) White | 1.00 |
| | 32 (8%) Black | 6 (8%) Black | 19 (9%) Black | |
| | 10 (2%) Other | 1 (0.01%) Other | 4 (0.02%) Other | |
| BMI (kg/m$^2$) | 26.4 ± 5.5 | 26.0± 5.3 | 26.9 ± 5.6 | 0.25 |
| **Admission diagnosis** | | | | |
| Concussion | 21 (5%) | 1 (1%) | 13 (6%) | 0.105 |
| Laceration/contusion | 35 (8%) | 5 (7%) | 23 (105) | 0.322 |
| SAH, SDH, extradural hemorrhage | 217 (50%) | 46 (61%) | 111 (50%) | 0.121 |
| Other unspecified ICH | 39 (9%) | 7 (9%) | 17 (8%) | 0.674 |
| Other intracranial injury | 119 (28%) | 17 (22%) | 57 (26%) | 0.549 |
| Tobacco use | 88 (23%) | 19 (26%) | 42 (20%) | 0.435 |
| Alcohol use | 155 (43%) | 25 (36%) | 84 (43%) | 0.430 |
| Recreational drug use | 23 (7%) | 5 (8%) | 8 (5%) | 0.346 |
| Depression history | 73 (19%) | 28 (39%) | 36 (17%) | < **0.001** |
| Anxiety history | 32 (8%) | 9 (13%) | 16 (7%) | 0.293 |
| NSAIDs use | 132 (44%) | 36 (47%) | 96 (43%) | 0.667 |
| Acetaminophen use | 59 (20%) | 43 (57%) | 16 (7%) | < **0.001** |
| Benzodiazepine use | 47 (16%) | 22 (29%) | 25 (11%) | < **0.001** |
| Muscle relaxant use | 24 (8%) | 12 (16%) | 12 (5%) | **0.009** |
| Antidepressants use | 72 (24%) | 27 (36%) | 45 (20%) | **0.013** |
| Antipsychotics use | 17 (6%) | 6 (8%) | 11 (5%) | 0.390 |
| **In-hospital exposures** | | | | |
| Intubation | 233/431 (54%) | 40/76 (53%) | 107/221 (48%) | 0.616 |
| Surgical Procedure | 119/297 (40%) | 34/76 (45%) | 81/221 (37%) | 0.211 |
| Intracranial surgery | 69/119 (58%) | 20/34 (59%) | 49/81 (60%) | 0.865 |
| Spine surgery | 32/119 (27%) | 9/34 (26%) | 23/81 (28%) | 0.834 |
| Other orthopedic surgery | 8/119 (7%) | 4/34 (12%) | 4/81 (5%) | 0.190 |
| Minor procedures | 10/119 (8%) | 1/34 (3%) | 9/81 (11%) | 0.156 |
| NSAIDS (# administrations) | 6.38 ± 9.34 [0.00, 98.00] | 6.01 ± 6.90 [0.00, 35.00] | 5.43 ± 7.32 [0.00, 73.00] | 0.666 |
| Benzodiazepines (# administrations) | 1.75 ± 3.41 [0.00, 30.00] | 1.82 ± 3.83 [0.00, 29.00] | 1.33 ± 3.26 [0.00, 30.00] | **0.002** |
| Muscle relaxants (# administrations) | 0.79 ± 3.09 [0.00, 36.00] | 1.66 ± 5.25 [0.00, 36.00] | 0.44 ± 1.81 [0.00, 13.00] | < **0.001** |
| Antidepressants (# administrations) | 2.90 ± 6.99 [0.00, 74.00] | 3.21 ± 6.44 [0.00, 33.00] | 2.32 ± 5.44 [0.00, 36.00] | < **0.001** |
| Opioids (# administrations) | 5.98 ± 7.86 [0.00, 92.00] | 5.66 ± 5.12 [0.00, 26.00] | 4.76 ± 5.94 [0.00, 37.00] | **0.003** |
| Average ME (mg/ (kg/m$^2$) per day) | 0.88 ± 2.19 [0.00, 21.07] | 0.83 ± 1.44 [0.00, 8.11] | 0.71 ± 1.92 [0.00, 21.07] | 0.612 |
| Opioid 48 hours prior to discharge | 199/431 (46%) | 63/76 (83%) | 136/221 (62%) | **0.001** |
| ICU length of stay (days) | 3.83 ± 9.58 [0.10, 179.00] | 2.80 ± 3.21 [0.30, 17.80] | 2.61 ± 3.49 [0.10, 31.00] | 0.676 |
| Hospital length of stay (days) | 9.28 ± 13.36 [0.30, 179.00] | 7.96 ± 8.02 [1.90, 49.90] | 7.68 ± 8.69 [0.30, 95.50] | 0.810 |
| Discharge | | | | |
| Home | 134/297 (45%) | 37/76 (51%) | 97/221 (44%) | 0.471 |
| Skilled Nursing | 72/297 (24%) | 16/76 (22%) | 56/221 (25%) | 0.453 |
| Rehabilitation | 62/297 (21%) | 14/76 (19%) | 48/221 (22%) | 0.542 |

*(Continued)*

**Table 1.** (Continued)

| Characteristic | Total (n = 431) | Preadmit Opioid Use (n = 76) | Opioid Naïve (n = 221) | p value |
|---|---|---|---|---|
| Death | 20/297 (7%) | 6/76 (8%) | 14/221 (6%) | 0.638 |

Data presented as number (percentage) or mean ± standard deviation [minimum, maximum].

Body mass index (BMI), intracranial hemorrhage (ICH), intensive care unit (ICU), morphine equivalent (ME), non-steroidal anti-inflammatory drug (NSAID), subarachnoid hemorrhage (SAH), subdural hemorrhage (SDH)

through each of these timepoints. Generalized linear models with a logit function were used to examine odd of persistent opioid prescription 1, 6 and 12 months after discharge (Table 3). Testing collinearity using variance inflation factors (VIF) indicated no problematic amount of collinearity among the independent variables (all VIF < 5). Preadmission opioid use was associated with increased odds of being prescribed opioids at 1 (OR 8.7, 95% CI 3.2–27.1, p<0.0001), 6 (OR 23.4, 95% CI 6.9–97.0, p<0.0001) and 12 months (OR 324.8, 95% CI 23.1–16907.5, p = 0.0004). Administration of opioids 48 hours prior to discharge (OR 3.3, 95% CI 1.4–8.0, p = 0.007) was associated with higher odds of being prescribed opioids 1-month post-

**Table 2. Descriptive statistics comparing post-discharge opioid use between preadmission opioid users and opioid naïve patients.**

| Medication | Total n = 431 n (%) | Preadmit opioid use n = 76 n (%) | Opioid naïve n = 221 n (%) | p value |
|---|---|---|---|---|
| **1 month** | | | | |
| Opioid use | 155 (52%) | 48 (87%) | 69 (41%) | < **0.001** |
| NSAIDs | 91 (30%) | 20 (36%) | 45 (27%) | 0.267 |
| Acetaminophen | 148 (50%) | 28 (51%) | 79 (48%) | 0.815 |
| Benzodiazepine | 50 (17%) | 17 (31%) | 23 (14%) | **0.009** |
| Muscle relaxants | 41 (14%) | 15 (27%) | 13 (8%) | < **0.001** |
| Antidepressants | 88 (29%) | 22 (40%) | 51 (31%) | 0.283 |
| Antipsychotics | 41 (14%) | 8 (15%) | 21 (13%) | 0.933 |
| **6 months** | | | | |
| Opioid use | 70 (35%) | 36 (69%) | 24 (23%) | < **0.001** |
| NSAIDs | 80 (40%) | 22 (42%) | 42 (40%) | 0.880 |
| Acetaminophen | 79 (39%) | 19 (37%) | 42 (40%) | 0.841 |
| Benzodiazepine | 30 (15%) | 11 (21%) | 14 (13%) | 0.292 |
| Muscle relaxants | 27 (13%) | 11 (21%) | 9 (9%) | **0.046** |
| Antidepressants | 72 (36%) | 22 (42%) | 39 (37%) | 0.620 |
| Antipsychotics | 21 (10%) | 8 (16%) | 8 (8%) | 0.195 |
| **12 months** | | | | |
| Opioid use | 47 (30%) | 22 (56%) | 17 (19%) | < **0.001** |
| NSAIDs | 71 (45%) | 19 (49%) | 41 (47%) | 1 |
| Acetaminophen | 60 (38%) | 17 (44%) | 33 (38%) | 0.687 |
| Benzodiazepine | 27 (17%) | 9 (23%) | 15 (17%) | 0.599 |
| Muscle relaxants | 18 (13%) | 8 (21%) | 8 (9%) | 0.089 |
| Antidepressants | 58 (37%) | 16 (41%) | 35 (40%) | 1 |
| Antipsychotics | 20 (13%) | 6 (15%) | 9 (10%) | 0.552 |

Data presented as number (percentage).

Non-steroidal anti-inflammatory drug (NSAID)

**Table 3. Generalized linear model comparing odds of remaining on opioids after discharge between preadmission opioid users and naive opioid patients.**

| Covariate | 1 month n = 117 | | 6 months n = 45 | | 12 months n = 21 | |
|---|---|---|---|---|---|---|
| | OR (95% CI) | p value | OR (95% CI) | p value | OR (95% CI) | p value |
| Preadmission opioid use | **8.68 (3.24–27.05)** | **<0.0001** | **23.44 (6.86–96.98)** | **< 0.0001** | **324.77 (23.10–16907.45)** | **0.0004** |
| Age | 0.99 (0.97–1.01) | 0.444 | **1.06 (1.02–1.11)** | **0.0046** | **1.14 (1.05–1.30)** | **0.012** |
| Sex (male) | 0.64 (0.30–1.37) | 0.251 | 0.92 (0.29–3.01) | 0.892 | 0.09 (0.01–0.87) | 0.053 |
| Race (non-white) | 1.18 (0.34–4.08) | 0.796 | 4.00 (0.76–22.41) | 0.105 | 1.22 (0.06–19.89) | 0.888 |
| Prior tobacco use | **0.18 (0.06–0.51)** | **0.002** | **0.21 (0.04–0.87)** | **0.037** | **0.03 (0.00–0.36)** | **0.014** |
| Prior alcohol use | 1.40 (0.6–3.05) | 0.391 | 1.39 (0.44–4.54) | 0.577 | 2.52 (0.21–36.47) | 0.462 |
| Prior recreational drug use | 1.34 (0.10–13.16) | 0.810 | 7.90 (0.64–191.27) | 0.144 | 0.52 (0.00–1504.57) | 0.831 |
| Prior benzodiazepine use | 1.00 (0.31–3.22) | 0.996 | 0.86 (0.17–4.71) | 0.853 | 3.82 (0.08–1399.21) | 0.552 |
| Prior antidepressant use | 1.30 (0.49–3.52) | 0.603 | 0.27 (0.05–1.29) | 0.100 | 4.01 (0.12–190.16) | 0.444 |
| Prior antipsychotic use | 2.23 (0.40–12.26) | 0.348 | 19.36 (0.66–2780.46) | 0.178 | 0.34 (0.00–449.00) | 0.685 |
| History of depression | 0.69 (0.24–1.93) | 0.477 | 5.10 (0.95–33.76) | 0.071 | 0.25 (0.01–8.21) | 0.419 |
| History of anxiety | 2.00 (0.44–9.15) | 0.365 | 0.55 (0.06–4.71) | 0.580 | 0.49 (0.02–13.12) | 0.659 |
| Inpatient intubation | 1.42 (0.62–3.33) | 0.417 | 2.05 (0.67–6.58) | 0.213 | 1.89 (0.24–16.07) | 0.535 |
| Hospital stay (days) | 0.98 (0.92–1.04) | 0.495 | 1.02 (0.96–1.11) | 0.687 | 0.99 (0.78–1.17) | 0.916 |
| Average ME (mg/(kg/m$^2$) per day) | 1.25 (0.94–1.84) | 0.186 | **1.80 (1.19–2.87)** | **0.008** | **4.48 (1.78–16.34)** | **0.006** |
| Opioids in last 48 hours prior to discharge | **3.27 (1.41–7.99)** | **0.007** | 3.50 (1.04–13.36) | 0.052 | 0.25 (0.01–3.05) | 0.292 |
| Discharge to skilled nursing facility | 1.79 (0.64–5.09) | 0.269 | 0.56 (0.11, 2.71) | 0.480 | **0.03 (0.00–0.61)** | **0.048** |
| Discharge to rehabilitation | 1.57 (0.60–4.24) | 0.361 | 2.25 (0.58, 9.38) | 0.248 | 1.48 (0.12–19.23) | 0.749 |

Data reported as odds ratio (95% confidence interval).

Body mass index (BMI), intensive care unit (ICU), non-steroidal anti-inflammatory drug (NSAID), morphine equivalent (ME).

discharge. Older age was associated with higher odds of persistent opioid prescription at 6 (OR 1.1, 95% CI 1.0–1.1, p = 0.005) and 12 months after discharge, as was increased average ME (day/kg /m$^2$) at 6 (OR 1.8, 95% CI 1.2–2.0, p = 0.008) and 12 months (OR 4.5, 95% CI 1.8–16.3, p = 0.006). History of tobacco use was associated with decreased risk of persistent opioid prescription at 1 (OR 0.2, 95% CI 0.1–0.5, p = 0.002), 6 (OR 0.2, 95% CI 0.04–0.9, p = 0.037) and 12 months (OR 0.03, 95% CI 0.00–0.36, p = 0.014). Discharge to a skilled nursing facility (OR 0.03, 90% CI 0.00–0.61, p = 0.048) was also associated with decreased risk.

## Comparison of surgical versus non-surgical patients

Generalized linear models were used to determine the influence of surgery on risk for persistent opioid prescription at 1, 6 and 12 months among patients with traumatic brain injury. Results in Table 4 show that there was a statistically significant main effect of pre-admission

**Table 4. Generalized linear model comparing the effect of preadmission opioid use and surgery on persistent opioid prescription 1, 6 and 12 months post-discharge.**

| Covariate | 1 month | | 6 months | | 12 months | |
|---|---|---|---|---|---|---|
| | OR (95% CI) | p value | OR (95% CI) | p value | OR (95% CI) | p value |
| Preadmission opioid use | **5.54 (2.18–16.12)** | **< 0.001** | **11.50 (4.25–33.74)** | **< 0.001** | **24.46 (6.60–199.91)** | **< 0.001** |
| Surgery | 0.97 (0.52–1.80) | 0.913 | 0.81 (0.26–2.39) | 0.709 | 0.64 (0.03–6.82) | 0.716 |
| Preadmission opioid use X surgery | 6.21 (0.86–127.91) | 0.116 | 1.30 (0.27–6.58) | 0.751 | 1.63 (0.11–42.85) | 0.725 |

opioid use on the likelihood of persistent opioid prescription at 1, 6, and 12 months; patients with pre-admission opioid use were more likely to be prescribed opioids at 1, 6, and 12 months. The main effect of surgery on the likelihood of chronic opioid use was not statistically significant; there were no statistically significant differences in the likelihood of chronic opioid use at 1, 6, and 12 months between surgical and non-surgical cases. The interaction effect between pre-admission opioid use and surgical cases was not statistically significant.

## Patients with missing opioid data

Among the 134 patients with no preadmission opioid data, post-discharge opioid usage data was missing for 55 patients at 1 month, 67 patients at 6 months and 69 patients respectively. Of those with valid post-discharge data, 38 (48%) were prescribed opioids 1 month post discharge; 7 (10%) were prescribed opioids at 6 months post-discharge and 3 (5%) were prescribed opioids at 12 months post-discharge. To evaluate for potential sources of bias, we compared preadmission characteristics of patients with and without preadmission opioid data. There were no statistically significant differences in sex (p = 0.22), race (p = 0.30), and BMI (p = 0.16). Patients with preadmission opioid data were statistically significantly older (mean age 65 years versus 57 years, p = 0.001), had a higher incidence of recreational drug use (10 (13%) versus 13 (5%), p = 0.04), a lower incidence of depression history (9 (9%) versus 64 (22%), p<0.001) and were more likely to be intubated (86 (64%) versus 147 (49%), p = 0.01).

## Discussion

Our study is the first to examine risk factors for persistent opioid prescription following ICU admission for TBI. A major finding of this study, which is of significant clinical importance, is that prior opioid use before an unanticipated ICU admission is associated with persistent opioid use up to a year after surgery. This is congruent with other perioperative studies investigating persistent opioid use after elective surgery. However, our findings are significantly different than Wang et al., who showed that amongst chronic opioid users, hospitalization with critical illness was not associated with substantial increases in opioids prescribed in the 6 months following hospitalization.[16] The implications of this on a practical level is improved vigilance, especially after discharge, of the high risk preoperative opioid user who sustains a TBI.

The amount of opioids patients receive during hospitalization (average OME per BMI per day) and opioid use prior to discharge (opioids in the last 48 hours) are significantly associated with persistent opioid prescription. This appears to be independent of injury severity as intubation, length of hospital stay, and discharge disposition were not associated with increased risk. These findings are similar to the work by Harbaugh et al, who showed that in patients undergoing third molar (wisdom tooth) extraction, persistent opioid use occurs at an adjusted rate of 13 (95% CI, 9–19) per 1000 patients who filled an opioid prescription compared with 5 (95% CI, 3–7) per 1000 patients who did not fill a perioperative prescription.[17] Our study is the first to demonstrate this important risk factor in a critical care cohort.

Together these findings suggest that ICU management of patients influences risk for persistent opioid prescription and may help identify patients at risk for chronic opioid use and misuse after hospital discharge. Reducing opioid exposure through use of non-opioid sedatives [18], multimodal analgesia [19] and limiting opioid prescription at discharge may play a key role in reducing the risk for chronic opioid dependence after TBI. This is consistent with the Society for Critical Care Medicine's 2018 Clinical Practice Guidelines for Management of Pain, Agitation/Sedation, Delirium, Immobility and Sleep Disruption which recommend systemic assessment with validated pain and sedation scales to reduce opioid use, use of

multimodal analgesia including non-opioid adjuncts and consideration of sedation agents, such as propofol or dexmedetomidine.[20]

TBI is a significant global health concern; however, few studies have investigated opioid use in this population. To date the majority of studies examining opioid use after TBI have focused on military service members.[21] This study extends these findings in a broader population group and identifies risk factors for persistent opioid prescription in this population.

Our study has several limitations: the study design was retrospective in nature with significant amounts of missing data. We characterized and compared the missing data with the complete cohort data, which demonstrated lower depression history and higher recreational drug use among patients with missing data. However, rates of chronic opioid prescription at 1, 6 and 12 months were similar and we do not believe that eliminating them from the analysis would have altered the results significantly.

Our cohort was intended to include only patients with a primary TBI; however, 40% of patients required a surgical intervention during the hospitalization, primarily intracranial or spine procedures. We do not believe that this significantly affected our results, as surgical cases were not associated with increased odds for persistent opioid use at 1, 6 or 12 months compared to non-surgical cases. The impact of poly-trauma requiring critical care needs to be investigated separately.

A significant limitation of our study is that we were unable to stratify patients by injury severity as measures of injury severity (i.e. Glasgow Coma Scale, Revised Trauma Score, APACHE score) are not routinely documented in our electronic medical record. We attempted to control for opioid exposure (mild injury/less exposure versus severe injury/ greater exposure) by normalizing the mg OME per BMI per day in our generalized in linear models. Whether or not a patient was intubated on admission, the length of hospital stay and discharge disposition were used as surrogates for injury severity, before admission, during hospitalization and after discharge, respectively. However, this represents a significant limitation of our study.

Finally, we observed that 44% of preadmission opioid users were not prescribed opioids 12 months after admission. Many of these patients were discharged to a rehabilitation or skilled nursing facility (57% (8/14) compared to 39% (7/18) preadmission opioid users who continued to be prescribed opioids). We were unable to reliably track opioid prescription as patients transitioned from in-patient hospitalization to SNF or rehabilitation care, thus the veracity of this data is in question. Interestingly, discharge to a skilled nursing facility was associated with decreased odds of persistent opioid prescription at 12 months. It is possible that patients discharged to SNF/rehabilitation facilities received opioids that were not documented in our study. The Virginia Prescription Monitoring Program could be used to validate opioid prescription data from multiple care facilities; however, this data is not available for research purposes.

Our results raise several important unanswered questions: Is decreasing opioid use in previous opioid requiring patients associated with worse modified Rankin Scale or Glasgow Coma Outcome Scales where opioids are not administered per patient request? Does re-establishing healthcare amongst previous opioid users following a TBI reduce opioid use? Our findings suggest areas for further exploration to determine factors associated with opioid cessation in this group of patients.

## Conclusions

Among primary TBI patients requiring critical care, history of opioid use, depression, higher average daily opioid dose during hospitalization, and receiving opioids 48h prior to discharge

are associated with increased risk of being prescribed opioids up to one year after discharge. These factors may be used to identify TBI patients who are at highest risk and target them for intervention and counseling.

## Supporting information

**S1 Table. Medication list.**
(DOCX)

**S2 Table. Average total morphine equivalent dose administered per day and week between preadmission opioid users and opioid naïve patients.**
(DOCX)

**S3 Table. Dataset.**
(CSV)

## Acknowledgments

The authors would like to thank Amir Abdel Malek, PhD, from the University of Virginia School of Medicine, Department of Anesthesiology, Charlottesville, VA for his assistance with data collection.

## Author Contributions

**Conceptualization:** Lauren K. Dunn, Joyce Chung, Marcel E. Durieux, Bhiken I. Naik.

**Data curation:** Davis G. Taylor, Samantha J Smith, Alexander J. Skojec, Tony R. Wang, Joyce Chung, Mark F. Hanak, Christopher D. Lacomis, Justin D. Palmer, Caroline Ruminski, Shenghao Fang, Sarah N. Spangler.

**Formal analysis:** Lauren K. Dunn, Siny Tsang.

**Investigation:** Lauren K. Dunn.

**Methodology:** Lauren K. Dunn, Siny Tsang, Bhiken I. Naik.

**Project administration:** Lauren K. Dunn.

**Supervision:** Lauren K. Dunn, Bhiken I. Naik.

**Validation:** Lauren K. Dunn, Siny Tsang, Sarah N. Spangler.

**Writing – original draft:** Lauren K. Dunn, Davis G. Taylor.

**Writing – review & editing:** Lauren K. Dunn, Davis G. Taylor, Samantha J Smith, Alexander J. Skojec, Tony R. Wang, Joyce Chung, Mark F. Hanak, Christopher D. Lacomis, Justin D. Palmer, Caroline Ruminski, Shenghao Fang, Siny Tsang, Sarah N. Spangler, Marcel E. Durieux, Bhiken I. Naik.

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
