## [Decision Letter · Decision Letter 0]

5 Jul 2019

PONE-D-19-15116

Persistent Post-Discharge Opioid Prescribing after Traumatic Brain Injury Requiring ICU Admission: A Cross-Sectional Study with Longitudinal Outcome

PLOS ONE

Dear Dr. Dunn,

Thank you for submitting your manuscript to PLOS ONE. After careful consideration, we feel that it has merit but does not fully meet PLOS ONE’s publication criteria as it currently stands. Therefore, we invite you to submit a revised version of the manuscript that addresses the points raised during the review process.

The reviewers suggest several modifications to the analysis that would improve its novelty, interpretation and alignment with your stated objectives. For example:

1. Please consider Reviewer 1's suggestions for additional analyses to explore how TBI is specifically a risk factor for prolonged opioid use.

2. Please address the concerns related to model overfitting

3. A major issue relates to the stated objective compared to the modeling approach taken (as described by the second reviewer). In particular, it appears that the models should be set-up with in-hospital opioid exposure as the main independent variable, with preadmission opioid use considered as an effect modifier (ie in a stratified analysis). 

We would appreciate receiving your revised manuscript by Aug 19 2019 11:59PM. To enhance the reproducibility of your results, we recommend that if applicable you deposit your laboratory protocols in protocols.io, where a protocol can be assigned its own identifier (DOI) such that it can be cited independently in the future. For instructions see: http://journals.plos.org/plosone/s/submission-guidelines#loc-laboratory-protocols

We look forward to receiving your revised manuscript.

Kind regards,

Tara Gomes

Academic Editor

PLOS ONE

Journal Requirements:

2.

We note that you have indicated that data from this study are available upon request. PLOS only allows data to be available upon request if there are legal or ethical restrictions on sharing data publicly. For information on unacceptable data access restrictions, please see http://journals.plos.org/plosone/s/data-availability#loc-unacceptable-data-access-restrictions.

Reviewers' comments:

Reviewer's Responses to Questions

**Comments to the Author**

1. Is the manuscript technically sound, and do the data support the conclusions?

Reviewer #1: No

Reviewer #2: Partly

2. Has the statistical analysis been performed appropriately and rigorously? 

Reviewer #1: Yes

Reviewer #2: No

3. Have the authors made all data underlying the findings in their manuscript fully available?

Reviewer #1: Yes

Reviewer #2: Yes

4. Is the manuscript presented in an intelligible fashion and written in standard English?

Reviewer #1: Yes

Reviewer #2: Yes

5. Review Comments to the Author

Reviewer #1: Please see uploaded review for :

Manuscript Number: PONE-D-19-15116

Article Type: Research Article

Full Title: Persistent Post-Discharge Opioid Prescribing after Traumatic Brain Injury Requiring

ICU Admission: A Cross-Sectional Study with Longitudinal Outcome

Short Title: Persistent Opioid Use After Traumatic Brain Injury

Corresponding Author: Lauren Kingsley Dunn

University of Virginia

Charlottesville, VA UNITED STATES

Keywords: traumatic brain injury; opioid dependence; opioid use disorder; Subarachnoid

hemorrhage; subdural hemorrhage

Reviewer #2: Thank you for the opportunity to review this interesting article. The authors performed a retrospective analysis of patients admitted to the neurological intensive care unit to evaluate risk factors for prolonged opioid use after discharge. They hypothesized that increased ICU opioid exposure would result in increased risk of prolonged opioid prescription. They separated their analysis into two groups: patients on pre-admission opioids and patients who were opioid naïve. They found that pre-admission opioid use and higher opioid requirements during hospitalization were associated with increased risk of being prescribed opioids 12 months post discharge.

The authors are to be commended on a well-written analysis of opioid prescribing patterns after admission for TBI. This is a timely topic, and of interest to readers. However, there are several areas that could be improved upon. Please see below for comments:

Page 6 – The authors lay out their argument for their interest in opioid use after TBI – 1) TBI is a significant global health concern with increasing incidence and significant cost associated 2) patients with TBI have been shown to have increased substance abuse disorders, including increased chronic use of opioids 3) however, opioids are ideal drugs to treat TBI in the acute phase 4) but they are also concerned that opioid use in the ICU will lead to prolonged prescription, as they themselves have already shown this to be a risk factor. Therefore 5) they want to see if exposure to opioids is a risk factor for prolonged prescription after TBI. It is unclear what new information about the topic they are offering, if they have already shown that TBI increases risk of chronic opioids, and opioid exposure in the hospital increases chronic opioids. Can the authors please more explicitly state what new information they are adding to the literature? (I.e., highlight the differences in you cohort – civilian vs military?, looking at only opioid-naïve patients?, etc.)

P9, L149 – Can the authors please comment on the reason for choosing the covariates?

P10, L183 – The model is confusing to me – why was a linear regression instead of logistic regression used for a dichotomous outcome? Opioid rx at 1,6,12 months (yes/no)? Additionally, considering only 48 patients still had an opioid rx at 12 months, the model is overfit with 19 covariates.

P14, L 222 – It might be reasonable to combine all the covariates of interest in Table 1. This would allow the authors to simplify the first paragraph on p14 into something more streamlined such as, “There were similar rates of intubation and surgery between the two groups. There was no difference in the types of surgeries between the two groups (see Table 1).” This will allow the text to be easier read and prevent redundant numbers.

P16, Table 2 – It is unclear why the model, which was supposed to determine whether the independent variable of ICU opioid-exposure is associated with the outcome, is presented in terms of pre-admission opioid users vs opioid-naïve patients? The question posited originally was “does ICU-opioid exposure increase prolonged opioid prescription?” Pre-admission opioid use should have been considered as a covariate in this model. In order to answer the original question, it seems the analysis should be redone, with the same outcome, the independent variable of ICU-opioid exposure, and only those covariates that influence the relationship between the independent variable and dependent variable, eg, pre-admission opioid exposure, without overfitting the model.

P 19 – Table 3-5 can be combined with columns for 1, 6, and 12 months for more expedient reading. This will also highlight easily across all time points that pre-admission opioid use was consistently significant.

P28, L350 – Is it not clear that in-hospital opioid exposure is independently associated with prolonged prescription. Presumably this effect is due to pre-admission opioid use, which led to higher inpatient use, which then led to continued post-ICU use. This should be addressed by either using an interaction term, or analyzing naïve and non-naïve patients separately to examine the effect of ICU-opioid exposure on outcome.

P28 Discussion – much of the introduction is repeated in the discussion. The authors have already made the case that opioid exposure after surgical procedures has been associated with increased risk of prolonged opioid use. A more effective use of this space would be to discuss how the author’s new data will influence practice. Is it feasible to avoid opioids in critically ill opioid-dependent patients? Are there other drugs that can be used in TBI care if you identify a patient to be at risk for prolonged prescription? How can one interpret the SCCM analagosedation guidelines given this data?

Minor:

Page 8, L 137 – Consistency with naming of the Neuro ICU – neurological (as in the abstract) or neuroscience?

P10, L186 – “Models were used to examine whether the odds of receiving opioid prescriptions at 1, 6, or 12 months after discharge…” This should be changed to AND to match the abstract and covariate section, which list the outcome as 1, 6, and 12 months. Otherwise the outcome is confusing, since the authors seem to be looking at a continuous exposure from 1-12months.

P14, line 222 – Intubation is not listed as one if the covariates on page 9.

6. PLOS authors have the option to publish the peer review history of their article (what does this mean?). If published, this will include your full peer review and any attached files.

Reviewer #1: No

Reviewer #2: No

---

## [Author Response · Author response to Decision Letter 0]

9 Sep 2019

Response to reviewers

The reviewers suggest several modifications to the analysis that would improve its novelty, interpretation and alignment with your stated objectives. 

We thank the Editor and Reviewers for the opportunity to submit a revised manuscript. We have attempted to address each of the reviewers concerns below. 

For example:

1.Please consider Reviewer 1's suggestions for additional analyses to explore how TBI is specifically a risk factor for prolonged opioid use.

Please see our response to Reviewer 1’s suggestions. Results of our generalized linear models show other factors associated with prolonged opioid use in this patient population, specifically in hospital opioid exposure, opioid administration < 48 prior to discharge. 

2. Please address the concerns related to model overfitting

To limit model overfitting, we reduced the number of covariates in each model by removing BMI, NSAIDs, acetaminophen, and muscle relaxants. In addition, we performed 3 separate analyses. 1) all covariates, 2) patient demographics and preadmission characteristics, 3) patient demographics and in hospital exposures. Results of models 2 and 3 were similar to model 1, thus we have included only model 1, with all the covariates in the current manuscript. Please see our response to Reviewer 2 comment 

3. A major issue relates to the stated objective compared to the modeling approach taken (as described by the second reviewer). In particular, it appears that the models should be set-up with in-hospital opioid exposure as the main independent variable, with preadmission opioid use considered as an effect modifier (ie in a stratified analysis). 

We respectfully note that Reviewer 2 was referring to Table 2 in the prior manuscript which displayed descriptive statistics. In-hospital opioid exposure (mg ME/kg/day) and preadmission opioid use are included as separate covariates in our generalized linear models shown in Tables 3 (1 month), 4 (6 months) and 5 (12 months). We have clarified this in the text. 

 We would appreciate receiving your revised manuscript by Aug 19 2019 11:59PM. When you are ready to submit your revision, log on to https://www.editorialmanager.com/pone/ and select the 'Submissions Needing Revision' folder to locate your manuscript file.  To enhance the reproducibility of your results, we recommend that if applicable you deposit your laboratory protocols in protocols.io, where a protocol can be assigned its own identifier (DOI) such that it can be cited independently in the future. For instructions see: http://journals.plos.org/plosone/s/submission-guidelines#loc-laboratory-protocols  Please include the following items when submitting your revised manuscript:

• A rebuttal letter that responds to each point raised by the academic editor and reviewer(s). This letter should be uploaded as separate file and labeled 'Response to Reviewers'.

• A marked-up copy of your manuscript that highlights changes made to the original version. This file should be uploaded as separate file and labeled 'Revised Manuscript with Track Changes'.

• An unmarked version of your revised paper without tracked changes. This file should be uploaded as separate file and labeled 'Manuscript'.

We look forward to receiving your revised manuscript.  Kind regards,  Tara Gomes Academic Editor PLOS ONE  Journal Requirements:

2.

We note that you have indicated that data from this study are available upon request. PLOS only allows data to be available upon request if there are legal or ethical restrictions on sharing data publicly. For information on unacceptable data access restrictions, please see http://journals.plos.org/plosone/s/data-availability#loc-unacceptable-data-access-restrictions.

Data has been made available.  

This phrase has been removed and appropriate data provided.

   Reviewers' comments:  Reviewer's Responses to Questions  Comments to the Author  1. Is the manuscript technically sound, and do the data support the conclusions?  The manuscript must describe a technically sound piece of scientific research with data that supports the conclusions. Experiments must have been conducted rigorously, with appropriate controls, replication, and sample sizes. The conclusions must be drawn appropriately based on the data presented.   Reviewer #1: No  Reviewer #2: Partly

 2. Has the statistical analysis been performed appropriately and rigorously?   Reviewer #1: Yes  Reviewer #2: No

 3. Have the authors made all data underlying the findings in their manuscript fully available?  The PLOS Data policy requires authors to make all data underlying the findings described in their manuscript fully available without restriction, with rare exception (please refer to the Data Availability Statement in the manuscript PDF file). The data should be provided as part of the manuscript or its supporting information, or deposited to a public repository. For example, in addition to summary statistics, the data points behind means, medians and variance measures should be available. If there are restrictions on publicly sharing data—e.g. participant privacy or use of data from a third party—those must be specified.  Reviewer #1: Yes  Reviewer #2: Yes

 4. Is the manuscript presented in an intelligible fashion and written in standard English?  PLOS ONE does not copyedit accepted manuscripts, so the language in submitted articles must be clear, correct, and unambiguous. Any typographical or grammatical errors should be corrected at revision, so please note any specific errors here.  Reviewer #1: Yes  Reviewer #2: Yes

 5. Review Comments to the Author  Please use the space provided to explain your answers to the questions above. You may also include additional comments for the author, including concerns about dual publication, research ethics, or publication ethics. (Please upload your review as an attachment if it exceeds 20,000 characters)  Reviewer #1: Please see uploaded review for :  Manuscript Number: PONE-D-19-15116 Article Type: Research Article Full Title: Persistent Post-Discharge Opioid Prescribing after Traumatic Brain Injury Requiring ICU Admission: A Cross-Sectional Study with Longitudinal Outcome Short Title: Persistent Opioid Use After Traumatic Brain Injury Corresponding Author: Lauren Kingsley Dunn University of Virginia Charlottesville, VA UNITED STATES Keywords: traumatic brain injury; opioid dependence; opioid use disorder; Subarachnoid hemorrhage; subdural hemorrhage 

The authors have chosen to review a timely and relevant issue, opioid exposure and the sequelae. Their aim, 

“to determine incidence and risk factors for persistent opioid prescription after hospitalization for TBI.”

The methodology employed was a retrospective review. The statistical analysis is noted to be appropriate.

There are however some limitations posed by the protocol design, the methodology and the conclusions, that must be noted/addressed:

#1. The premise of the study is an extrapolation of risk from the military TBI cohort. This is however difficult, as the TBI military cohort, also known as the Polytrauma cohort does not have a high specificity in identifying TBI. For many reasons, patients are included as TBI, allowing a high sensitivity in the diagnosis at the cost of specificity. These patients also have significant co-morbid injuries that impact comparisons of risk for opioid use. Thus any extrapolation is significantly limited.

We thank Reviewer 1 for this comment. It was not our intention to compare risk for opioid use in our TBI cohort to previous studies in military veterans. Military veteran data were discussed only to highlight that prior studies have focused only on this cohort and data regarding risk for opioid use in the general population after TBI is lacking, and as the Reviewer notes represents a very different group. We have limited discussion of the military veteran population to the discussion.

#2. The exclusion of major injuries is noted, 

“to remove the confounding effect of non- neurological injuries on persistent post discharge opioid prescription…” 

However, there is no confirmation that these patients are isolated TBI. There are many other conflating injuries that might significantly impact pain and subsequent opioid use. These are not noted to be excluded. Further supporting this concern, is that spine surgical patients are included in the cohort. 

No injuries outside of intracranial and spinal injuries in this ICU. Sensitivity analysis removing patients having surgical procedure.

We thank Reviewer #1 for making this important point. Our patient cohort was selected based on admission to our Neuroscience Intensive Care Unit which cares for patients with isolated traumatic brain injury or spine injury. Patients presenting with polytrauma are admitted to our Surgical Trauma Intensive Care Unit. This has been added to the methods page 5 line 104-106. 

Number of surgical procedures by type has been added to Table 1. Of 115 surgical procedures, a majority were intracranial or spine procedures. Orthopedic fractures and other minor surgical procedures (skin debridement, tracheostomy) accounted for <10% each. 

We have also included comparison of the effect of surgery on risk for persistent opioid use in the Results page 19, lines 511-520. “There was a statistically significant effect of pre-admission opioid use on the likelihood of persistent opioid prescription at 1, 6, and 12 months; patients with pre-admission opioid use were more likely to be prescribed opioids at 1, 6, and 12 months. The main effect of surgery on the likelihood of chronic opioid use was not statistically significant; there were no statistically significant differences in the likelihood of chronic opioid use at 1, 6, and 12 months between surgical and non-surgical cases. The interaction effect between pre-admission opioid use and surgical cases was not statistically significant.”

#3. The TBI patients are not stratified by severity. 

“Occasionally high doses of opioids are employed for sustained periods of time in patients with a severe TBI requiring mechanical ventilation…”

Severely injured patients are alluded to as needing sedation as a risk factor, but without stratification, it is unclear how the identified risk factors impacts the specific cohorts. As an example, mild patients would have less exposure and presumably less risk????

Again, Reviewer #1 makes an excellent point. Unfortunately, measures of injury severity (i.e. Glasgow Coma Scale, Revised Trauma Score, APACHE score) are not routinely documented in our electronic medical record. We attempt to control for opioid exposure (mild injury/less exposure vs severe injury/greater exposure) by normalizing mg oral morphine equivalent per BMI per day in our generalized in linear models. In addition, we included 3 covariates in our generalized linear models 1) whether or not a patient was intubated on admission, 2) length of hospital stay and 3) discharge disposition as surrogates for injury severity before admission, during hospitalization and after discharge, respectively. We have emphasized this a limitation in the discussion section Page 23 lines 699-706. 

#4. The stratification of patients with pre versus post injury opioid use is of benefit. The findings however do not differ from the established understanding of risk associated with previous use. Unclear how TBI becomes an additional risk factor beyond previous use.

Preadmission opioid use is one factor associated with risk for chronic opioid prescription. Key findings from our study are that, after controlling for measures of injury severity and hospital length of stay, the amount of opioids that patients are exposed to during their hospitalization and whether or not they are weaned off opioid prior to discharge is significantly associated with their risk for being prescribed opioids one year after discharge. This suggests that ICU management of patients may influence their risk for chronic opioid use. Strategies aimed at reducing opioid exposure, such as use of non-opioid sedative infusions and multimodal analgesia and limiting opioid prescription at discharge may play a key role in reducing the risk for chronic opioid dependence after TBI and warrant future study. We clarify the importance of these findings in our discussion. Please also see our response to Reviewer 2 comment 1. 

#5. The discussion section focuses on the perils of opioid use established, but fails to either clearly establish the risks from the study nor the link to the devastating sequelae highlighted.

Please see our response to Comment #4 above. We have revised the discussion section in order to more clearly highlight the important findings from our study which are that in-hospital management of these patient may contribute to long-term risk for chronic opioid prescription. Potential risk reduction strategies such as use of non-opioid sedative agents (i.e. dexmedetomidine) or multimodal analgesia and limiting opioid prescribing at discharge are important areas for future study. 

In summary, it is agreed that the risk associated with opioid use in TBI patients is of concern. The manuscript however does not clearly identify risk factors specific to TBI beyond the known pre- injury use and further does not account for concomitant injuries.

Would recommend a cohort of isolated TBI versus those with additional injuries to better understand opioid use in TBI.

We thank the Reviewer for this recommendation. Please see our response to Comment #1. We hope that the inclusion of an analysis of the effect of surgery on risk for persistent opioid use appropriately addresses this concern. 

Would also recommend stratification of injury type, for better understanding of the risk assumed to exist in the severe cohort.

We sincerely thank the Reviewer for this recommendation. Unfortunately, we are limited in our ability to stratify patients by injury severity as the necessary measures (GCS, Revised trauma score, APACHE score) are not documented in our electronic medical record. We recognize that this is a significant limitation of our study, which we have attempted to clearly articulate in the discussion section. We control for opioid exposure by normalizing for mg ME per BMI per day and account for injury severity to the best of our ability by using surrogate measures (intubation, length of stay, and discharge disposition) in our models. 

 Reviewer #2: Thank you for the opportunity to review this interesting article. The authors performed a retrospective analysis of patients admitted to the neurological intensive care unit to evaluate risk factors for prolonged opioid use after discharge. They hypothesized that increased ICU opioid exposure would result in increased risk of prolonged opioid prescription. They separated their analysis into two groups: patients on pre-admission opioids and patients who were opioid naïve. They found that pre-admission opioid use and higher opioid requirements during hospitalization were associated with increased risk of being prescribed opioids 12 months post discharge.  The authors are to be commended on a well-written analysis of opioid prescribing patterns after admission for TBI. This is a timely topic, and of interest to readers. However, there are several areas that could be improved upon. Please see below for comments:  Page 6 – The authors lay out their argument for their interest in opioid use after TBI – 1) TBI is a significant global health concern with increasing incidence and significant cost associated 2) patients with TBI have been shown to have increased substance abuse disorders, including increased chronic use of opioids 3) however, opioids are ideal drugs to treat TBI in the acute phase 4) but they are also concerned that opioid use in the ICU will lead to prolonged prescription, as they themselves have already shown this to be a risk factor. Therefore 5) they want to see if exposure to opioids is a risk factor for prolonged prescription after TBI. It is unclear what new information about the topic they are offering, if they have already shown that TBI increases risk of chronic opioids, and opioid exposure in the hospital increases chronic opioids. Can the authors please more explicitly state what new information they are adding to the literature? (I.e., highlight the differences in you cohort – civilian vs military?, looking at only opioid-naïve patients?, etc.)

Thank you for the succinct overview of the study. 

1. One of the major findings of this study, which is of significant clinical importance, is that prior opioid use before an unanticipated ICU admission is associated with persistent opioid use up to a year after surgery. This finding is congruent with other perioperative studies investigating persistent opioid use after elective surgery. However, our findings are significantly different than Wang et al. (Crit Care Med. 2018 Dec;46(12):1934-1942), who showed that amongst chronic opioid users, hospitalization with critical illness was not associated with substantial increases in opioids prescribed in the 6 months following hospitalization. The implications of this finding on a practical level is improved vigilance, especially after discharge, of the high risk preoperative opioid user who sustains a TBI.

2. A second important finding of our study is the significance of opioid use prior to discharge (Opioids last 48 hrs). This finding is similar to the work by Harbaugh et al (JAMA. 2018 Aug 7;320(5):504-506) who showed that in patients undergoing third molar (wisdom tooth) extraction, persistent opioid use occurs at an adjusted rate of 13 (95% CI, 9-19) per 1000 patients who filled an opioid prescription compared with 5 (95% CI, 3-7) per 1000 patients who did not fill a perioperative prescription. Our study is the first to demonstrate this important risk factor in a critical care cohort. 

We have limited our discussion of the military cohort in the introduction and revised the discussion to highlight what new information our study is providing. Please also see our response to Reviewer 1 comment 4. 

 P9, L149 – Can the authors please comment on the reason for choosing the covariates?

Covariates were chosen a priori based on previous studies that demonstrated an association with persistent opioid use after surgery (e.g. antidepressant use, substance abuse history), biological plausibility (e.g. in-hospital opioid exposure) and previously unexplored variables (e.g. opioid exposure 48 hrs prior to discharge and discharge disposition). This has been added to the Methods Page 6, lines 128-132.   P10, L183 – The model is confusing to me – why was a linear regression instead of logistic regression used for a dichotomous outcome? Opioid rx at 1,6,12 months (yes/no)? Additionally, considering only 48 patients still had an opioid rx at 12 months, the model is overfit with 19 covariates.

In response to your first question on linear vs. logistic regression:

1. P10 L183 refers to the models used to examine differences between opioid naïve and chronic opioid patents. As described in P10 L185, we use generalized linear models to examine whether the odds of receiving opioid prescriptions (yes/no) differ between opioid naïve and chronic opioid patients. When the outcome variable is dichotomous, generalized linear models with a logit function can also be referred as logistic regression models. In the revised manuscript, we clarified that we used a logit function in our generalized linear models.

With regard to the question of over-fitting, the reviewer makes an important observation which we have addressed in the following way:

1. We have reduced the number of co-variates in our full model for each time period by removing BMI, muscle relaxants, NSAIDS, and Acetaminophen

2. To further reduce the risk of overfitting we created 2 additional models for the three time points (1, 6 and 12 months). They include:

a. Demographic and Preadmission Exposures

b. Demographics + In-hospital Exposures

Results of the two additional models were similar so we elected to include only the results of the full model for each time point in Table 3. Please see results of these additional models below: 

Persistent Opioid Prescription (1 month)

Full model (1 month)

Demographics and preadmit exposures (1 month):

Demographics and in hospital exposures (1 month)

\f

Persistent Opioid Prescription (6 month)

Full model (6 month)

Demographics and preadmit exposures (6 month):

\f

Demographics and in hospital exposures (6 month)

\f

Persistent Opioid Prescription (12 month)

Full model (12 month)

Demographics and preadmit exposures (12 month):

Demographics and in hospital exposures (12 month)

  

\f

P14, L 222 – It might be reasonable to combine all the covariates of interest in Table 1. This would allow the authors to simplify the first paragraph on p14 into something more streamlined such as, “There were similar rates of intubation and surgery between the two groups. There was no difference in the types of surgeries between the two groups (see Table 1).” This will allow the text to be easier read and prevent redundant numbers.

Thank you for this suggestion. We have made the appropriate changes in the manuscript and added In-hospital Characteristics and Exposures to Table 1 

 P16, Table 2 – It is unclear why the model, which was supposed to determine whether the independent variable of ICU opioid-exposure is associated with the outcome, is presented in terms of pre-admission opioid users vs opioid-naïve patients? The question posited originally was “does ICU-opioid exposure increase prolonged opioid prescription?” Pre-admission opioid use should have been considered as a covariate in this model. In order to answer the original question, it seems the analysis should be redone, with the same outcome, the independent variable of ICU-opioid exposure, and only those covariates that influence the relationship between the independent variable and dependent variable, eg, pre-admission opioid exposure, without overfitting the model.

Table 3 on Page 16 (Table 2 in revised manuscript) is a univariate analysis on post-discharge medications between patients with Preadmit Opioid Use vs. Opioid Naïve. There is no multivariable modeling reported in this table. Preadmission opioid use is used as a covariate in all our multivariable modeling (current Table 3). As discussed earlier, to reduce overfitting we have provided additional restricted models. 

 P 19 – Table 3-5 can be combined with columns for 1, 6, and 12 months for more expedient reading. This will also highlight easily across all time points that pre-admission opioid use was consistently significant.

Done  P28, L350 – Is it not clear that in-hospital opioid exposure is independently associated with prolonged prescription. Presumably this effect is due to pre-admission opioid use, which led to higher inpatient use, which then led to continued post-ICU use. This should be addressed by either using an interaction term, or analyzing naïve and non-naïve patients separately to examine the effect of ICU-opioid exposure on outcome.

We respectfully disagree with the reviewer regarding his concern about in-hospital opioid exposure correlating to preadmission opioid use. For each full model we tested for collinearity using variance inflation factors (VIF) which indicated no problematic amount of collinearity among the independent variables (all VIF < 5) for Average ME (day/kg/m2) and Chronic opioid user. This has been noted on page 15, lines 326-328.   P28 Discussion – much of the introduction is repeated in the discussion. The authors have already made the case that opioid exposure after surgical procedures has been associated with increased risk of prolonged opioid use. A more effective use of this space would be to discuss how the author’s new data will influence practice. Is it feasible to avoid opioids in critically ill opioid-dependent patients? Are there other drugs that can be used in TBI care if you identify a patient to be at risk for prolonged prescription? How can one interpret the SCCM analagosedation guidelines given this data?

Thank you for this suggestion. We have modified our discussion based on your excellent suggestion (Page 21-22, lines 577-588).     Minor:  Page 8, L 137 – Consistency with naming of the Neuro ICU – neurological (as in the abstract) or neuroscience?

This has been corrected.   P10, L186 – “Models were used to examine whether the odds of receiving opioid prescriptions at 1, 6, or 12 months after discharge…” This should be changed to AND to match the abstract and covariate section, which list the outcome as 1, 6, and 12 months. Otherwise the outcome is confusing, since the authors seem to be looking at a continuous exposure from 1-12months.

This has been corrected.  P14, line 222 – Intubation is not listed as one if the covariates on page 9.

This has been added.

---

## [Decision Letter · Decision Letter 1]

4 Nov 2019

PONE-D-19-15116R1

Persistent post-discharge opioid prescribing after traumatic brain injury requiring intensive care unit admission: a cross-sectional study with longitudinal outcome

PLOS ONE

Dear Dr. Dunn,

Thank you for submitting your manuscript to PLOS ONE. After careful consideration, we feel that it has merit but does not fully meet PLOS ONE’s publication criteria as it currently stands. Therefore, we invite you to submit a revised version of the manuscript that addresses the points raised during the review process.

We would appreciate receiving your revised manuscript by Dec 19 2019 11:59PM. To enhance the reproducibility of your results, we recommend that if applicable you deposit your laboratory protocols in protocols.io, where a protocol can be assigned its own identifier (DOI) such that it can be cited independently in the future. For instructions see: http://journals.plos.org/plosone/s/submission-guidelines#loc-laboratory-protocols

We look forward to receiving your revised manuscript.

Kind regards,

Tara Gomes

Academic Editor

PLOS ONE

Reviewers' comments:

Reviewer's Responses to Questions

**Comments to the Author**

1. If the authors have adequately addressed your comments raised in a previous round of review and you feel that this manuscript is now acceptable for publication, you may indicate that here to bypass the “Comments to the Author” section, enter your conflict of interest statement in the “Confidential to Editor” section, and submit your "Accept" recommendation.

Reviewer #2: (No Response)

2. Is the manuscript technically sound, and do the data support the conclusions?

Reviewer #2: Yes

3. Has the statistical analysis been performed appropriately and rigorously? 

Reviewer #2: Yes

4. Have the authors made all data underlying the findings in their manuscript fully available?

Reviewer #2: Yes

5. Is the manuscript presented in an intelligible fashion and written in standard English?

Reviewer #2: Yes

6. Review Comments to the Author

Reviewer #2: The authors have done a good job responding to most of my concerns. The table updates make the article easier to read. However, there is an outstanding issues:

My confusion regarding the modeling can be clarified if the introduction is changed. While I still do not think the current model answers their stated hypothesis (ICU opioid exposure as an independent variable affecting outcome) – it does answer a question. It appears that the authors are just searching for risk factors for persistent opioid prescription in patients with TBI. However, they couch all their data and discussion in terms of pre-admission opioid users and opioid naïve patients. The hypothesis in the introduction could be changed to the following:

“The aim of this study is to determine the incidence and risk factors for persistent opioid prescription (up to one year after admission) in patients with a primary TBI. We hypothesized that opioid use prior to hospitalization and in-hospital exposure to opioids for management of TBI would be associated with increased risk for persistent opioid prescription 1 year after hospital discharge.”

This would clarify why they present all the data stratified by pre-admission exposure vs naïve.

With this, it should be ready for publication from my standpoint.

7. PLOS authors have the option to publish the peer review history of their article (what does this mean?). If published, this will include your full peer review and any attached files.

Reviewer #2: No

---

## [Author Response · Author response to Decision Letter 1]

6 Nov 2019

Response to Reviewers

Reviewer #2: The authors have done a good job responding to most of my concerns. The table updates make the article easier to read. However, there is an outstanding issue:  My confusion regarding the modeling can be clarified if the introduction is changed. While I still do not think the current model answers their stated hypothesis (ICU opioid exposure as an independent variable affecting outcome) – it does answer a question. It appears that the authors are just searching for risk factors for persistent opioid prescription in patients with TBI. However, they couch all their data and discussion in terms of pre-admission opioid users and opioid naïve patients. The hypothesis in the introduction could be changed to the following:  “The aim of this study is to determine the incidence and risk factors for persistent opioid prescription (up to one year after admission) in patients with a primary TBI. We hypothesized that opioid use prior to hospitalization and in-hospital exposure to opioids for management of TBI would be associated with increased risk for persistent opioid prescription 1 year after hospital discharge.” This would clarify why they present all the data stratified by pre-admission exposure vs naïve.  With this, it should be ready for publication from my standpoint.

We thank Reviewer 2 for their review of our revise manuscript. We have made the recommended change to the hypothesis stated in the Introduction Page 4, line 66-70.

---

## [Editor Report · Decision Letter 2]

13 Nov 2019

Persistent post-discharge opioid prescribing after traumatic brain injury requiring intensive care unit admission: a cross-sectional study with longitudinal outcome

PONE-D-19-15116R2

Dear Dr. Dunn,

We are pleased to inform you that your manuscript has been judged scientifically suitable for publication and will be formally accepted for publication once it complies with all outstanding technical requirements.

With kind regards,

Tara Gomes

Academic Editor

PLOS ONE
---

## [Editor Report · Acceptance letter]

18 Nov 2019

PONE-D-19-15116R2 

Persistent post-discharge opioid prescribing after traumatic brain injury requiring intensive care unit admission: a cross-sectional study with longitudinal outcome 

Dear Dr. Dunn:

I am pleased to inform you that your manuscript has been deemed suitable for publication in PLOS ONE. Congratulations! Your manuscript is now with our production department. 

With kind regards,

on behalf of

Dr. Tara Gomes 

Academic Editor

PLOS ONE